# Differences in the Prevalence and Profile of DSM-IV and DSM-5 Alcohol Use Disorders—Results from the Singapore Mental Health Study 2016

**DOI:** 10.3390/ijerph20010285

**Published:** 2022-12-24

**Authors:** Mythily Subramaniam, Edimansyah Abdin, Alexander Man Cher Kong, Janhavi Ajit Vaingankar, Anitha Jeyagurunathan, Saleha Shafie, Rajeswari Sambasivam, Daniel Shuen Sheng Fung, Swapna Verma, Siow Ann Chong

**Affiliations:** 1Research Division, Institute of Mental Health, Singapore 539747, Singapore; 2Saw Swee Hock School of Public Health, National University of Singapore, Singapore 117549, Singapore; 3Duke-NUS Medical School, Singapore 169857, Singapore; 4Medical Board, Institute of Mental Health, Singapore 539747, Singapore

**Keywords:** alcohol use disorder, population survey, Asian, multi-ethnic, comorbidity

## Abstract

Introduction: The Diagnostic and Statistical Manual of Mental Disorders, 5th edition (DSM-5) criteria for alcohol use disorders (AUD) was a significant shift from the historical DSM-IV Text Revised version. Following this shift in diagnostic criteria, a difference in the prevalence of AUD was expected. The current study aimed to evaluate the consequences of the modification of the diagnostic criteria from DSM-IV to DSM-5 AUD using lifetime diagnosis in Singapore’s multi-ethnic population using data from a nationwide epidemiological study. Methods: Respondents were assessed for lifetime mental disorders using the Composite International Diagnostic Interview (CIDI) administered through face-to-face interviews. Lifetime DSM-IV AUD diagnoses were compared with DSM-5 AUD diagnoses generated by modifying the criteria and the addition of the craving criterion. Sociodemographic correlates of lifetime DSM-IV/DSM-5 AUD were examined using multiple logistic regression analysis. Associations between DSM-IV/DSM-5 AUD, chronic conditions, and the HRQOL summary scores were examined using logistic and linear regression after controlling for significant sociodemographic factors. Results: The lifetime prevalence of DSM-IV AUD was 4.6% (0.5% for dependence and 4.1% for abuse) in the adult population, while the lifetime prevalence of DSM-5 AUD was 2.2%. Younger age, male gender, and lower education were associated with higher odds of both DSM-IV and DSM-5 AUD. However, those who were economically inactive (versus employed) (AOR, 0.4; 95% CI 0.2–0.9) and had a higher monthly household income (SGD 4000–5999 versus below SGD 2000) had lower odds of DSM-IV AUD (AOR, 0.4; 95% CI 0.2–0.7), but this was not observed among those with DSM-5 AUD. Both DSM-IV and DSM-5 AUD were associated with significant comorbidities in terms of other mental disorders; however, those diagnosed with lifetime GAD had significantly higher odds of having DSM-5 AUD (AOR, 5; 95% CI 1.9–13.2) but not DSM-IV AUD. Conclusions: While using the DSM-5 criteria would result in a lower prevalence of AUD in Singapore, it remains a highly comorbid condition associated with a poor health-related quality of life that is largely untreated, which makes it a significant public health concern.

## 1. Introduction

Alcohol use has become embedded in social practices globally, with moderate use shown to enhance positive affect and bonding [1]. On the other hand, the harmful use of alcohol is a leading public health concern worldwide that results in millions of deaths consequent to non-communicable diseases and injuries. Globally in 2016, there were 99.2 million disability-adjusted life-years (DALYs), and 4.2% of all DALYs were attributable to alcohol use [2]. Alcohol-related conditions result in a significant economic burden due to the loss of productivity as well as the cost of treatment of alcohol-related disorders and injuries. A systematic review conducted across European countries indicated that the social cost of alcohol per-capita in 2014 ranged from €26 to €1500 [3].

The Diagnostic and Statistical Manual of Mental Disorders, 5th edition (DSM-5) criteria for AUD [4] was a significant shift from the historical DSM-IV Text Revised version [5] of a binary construct of abuse and dependence as two mutually distinct conditions to a unitary construct of AUD with a continuum of severity. This revision aimed to overcome the problems associated with the DSM-IV classification, including poor reliability and validity of abuse when assessed hierarchically compared to dependence, which improved when the abuse criteria were analyzed without regard to dependence [6,7].

Other significant changes in the fifth edition of the DSM include removal of the criterion related to alcohol-related legal problems, the addition of a new craving criterion, and specifying AUD severity as mild, moderate, or severe based on the number of diagnostic criteria endorsed (2–3 symptom criteria indicating mild AUD, 4–5 symptom criteria indicating moderate AUD, and 6+ symptom criteria indicating severe AUD). Thus, at least two out of the total 11 criteria are required for a diagnosis of AUD. This contrasted with the DSM-IV alcohol abuse diagnosis, which required the endorsement of one out of four symptom criteria, and alcohol dependence, which required three out of seven symptom criteria.

Following this shift in diagnostic criteria, a difference in the prevalence of AUD was expected [8]. While these changes imply that all individuals with DSM-IV alcohol dependence will meet the DSM-5 AUD diagnostic threshold, those who had met DSM-IV abuse diagnosis based on one criterion would not meet the criteria for DSM-5 AUD. On the other hand, some of the “diagnostic orphans” [9,10], i.e., those meeting only two dependence criteria and none for abuse and, therefore, not diagnosed as DSM-IV AUD, would be positive for DSM-5 AUD. Bartoli, Carra, Crocamo, and Clerici [11] conducted a review of 12 studies. The majority of these were from the United States of America (USA), while the rest were from Europe, and one study was from Israel. Seven of the studies included in the review showed an increase, two found no substantial difference, while three studies found a decrease in the prevalence of AUD according to the DSM-5 diagnostic criteria. The authors concluded that the DSM-5 diagnostic criteria “inflated prevalence rates of AUD,” especially in the non-clinical populations. They attributed this increase to ‘diagnostic orphans’ being more prevalent than those meeting the single criterion of DSM-IV alcohol abuse. However, Slade et al. [12], using cross-national data from nine countries that participated in the World Health Organization’s (WHO) World Mental Health Survey (WMHS; countries were from South America, Europe, and the Middle East (Iraq)), concluded that the prevalence of DSM-5 AUD was 12% lower than that of DSM-IV AUD. In the pooled sample, the lifetime prevalence of DSM-IV AUD was 12.3% (8.0% abuse, 4.3% for dependence), while the prevalence of DSM-5 AUD was 10.8% (5.6% mild, 2.3% moderate, and 2.8% severe). The authors concluded that the difference was due to more abuse cases (meeting only one criterion) than the diagnostic orphans.

Singapore is a Southeast Asian country with a multi-ethnic population of approximately 5.7 million people [13]. Singapore has a low prevalence of binge drinking and AUD as compared to countries worldwide [14,15]. The overall low prevalence of alcohol consumption has also been attributed to several public health measures in Singapore, including the high taxes on alcohol imports. According to the GoEuro 2016 beer price index that ranks 70 cities according to the cost of beer, Singapore was the third most expensive city to buy beer. A bill passed in 2015 (Liquor Control (Supply and Consumption) Bill) banned public consumption of alcohol from 10:30 pm to 7 am in public places island-wide, and retails shops were prohibited from selling liquor after 10:30 pm. Therefore, the authors were interested in understanding the consequences of the changed criteria given the background of relatively low consumption, legal strictures, and ethnic differences in the population.

The current study attempts to evaluate the consequences of the modification of the diagnostic criteria from DSM-IV to DSM-5 AUD using lifetime diagnosis in Singapore’s multi-ethnic population using data from a nationwide epidemiological study. The aims of the current study were thus to (i) compare the differences in the lifetime prevalence of DSM-IV and DSM-5 AUD in a nationally representative sample of adults in Singapore and (ii) to examine the sociodemographic correlates, comorbidity with psychiatric and physical disorders, and health-related quality of life (HRQOL) among individuals meeting the lifetime DSM-IV and DSM-5 AUD diagnoses.

## 2. Materials and Methods

### 2.1. Sample

The Singapore Mental Health Study 2016 (SMHS 2016) was a population-based, cross-sectional epidemiological study. Singapore citizens and permanent residents aged 18 years and above living in Singapore during the study field period were included in the study. A disproportionate stratified probability sample was randomly selected using a sampling frame comprising an administrative database of residents in Singapore. Residents aged 65 and above, Malays, and Indians were oversampled to ensure that a sufficient sample size would be achieved. The study has been described in greater detail in an earlier article [16].

The study was approved by the institutional ethics committee (National Healthcare Group, Domain Specific Review Board, Singapore), and all participants gave written informed consent before initiation of the study procedures. Additionally, parental consent was sought for participants below 21 years of age. Six thousand one hundred twenty-six respondents agreed to participate in the study, which yielded a response rate of 69.5% for the study.

### 2.2. Measures—World Health Organization’s Composite International Diagnostic Interview Version 3.0

The diagnosis of mental disorders was established using the World Health Organization’s Composite International Diagnostic Interview (CIDI) Version 3.0 [17]. Mood disorders assessed in the SMHS 2016 were major depressive disorder (MDD) and bipolar disorder. Anxiety disorders included generalized anxiety disorder (GAD) and obsessive- compulsive disorder (OCD). The CIDI organic exclusion rules were applied to all diagnoses. Lifetime suicidality, defined as any suicidal ideation, suicide plan, or suicide attempt, was assessed using the ‘Suicidality’ module of the CIDI 3.0.

To assess for DSM-IV AUD, the participants were asked how old they were when they first consumed at least 12 alcohol drinks (either a glass of wine, a can or bottle of beer, or a shot or jigger of liquor either alone or in a mixed drink) in any 1 year in their lifetime. The individuals who responded ‘no’ to this question were asked a few additional questions and then ‘skipped out’ of the alcohol section. The individuals who responded ‘yes’ were asked a series of questions on their alcohol use. Those who answered positively were administered the symptom questions that operationalized the DSM-IV AUD diagnostic criteria. These questions comprised four alcohol abuse criteria (failure to fulfill major role, legal problems, use in hazardous situations, and social or interpersonal problems) and seven alcohol dependence criteria (tolerance, withdrawal, use more than intended, inability to cut down, larger time spent obtaining or recovering, other pursuits given up, and continued use despite knowledge of harm).

To assess the DSM-5 craving criterion, the endorsement of AU 19 (‘wanted a drink so badly you couldn’t think of anything else?’) in the Alcohol Use section of the CIDI 3.0 was used. The legal criterion from the DSM IV was excluded, and the endorsement of the remaining criteria from DSM IV alcohol abuse and dependence in the CIDI 3.0 was used to assess for DSM-5 AUD.

A treatment gap was ascertained by asking the respondents whether they had ever in their life ‘talked to a medical doctor or other professional (psychologists, counsellors, spiritual advisors, herbalists, acupuncturists, and any other healing professionals)’ about their problems associated with the use of alcohol. Those who had never sought treatment for their alcohol-related problems were assessed as having a treatment gap.

### 2.3. Other Questionnaires

The Modified Fagerstrom test for Nicotine Dependence [18] was administered to current smokers to establish nicotine dependence. Those with scores five and above were categorized as dependent as defined by previous studies [19].

Chronic medical conditions were assessed using a modified version of the World Health Organization’s Composite International Diagnostic Interview (CIDI) version 3.0 [20]. The question reads as “I am going to read to you a list of health problems some people have. Has a doctor ever told you that you have any of the following chronic medical conditions?” This was followed by a list of 18 chronic physical conditions considered prevalent in Singapore.

Quality of life was assessed using the Short Form (SF)-12 instrument that is designed to measure the generic quality of life [21]. SF-12 generates scores across eight domains of health, which are used to generate two summary scores: physical component summary (PCS) and mental component summary (MCS). The summary scores range between 0 and 100 with higher scores indicating better health.

The sociodemographic information obtained included the age at interview (18–34, 35–49, 50–64, or 65 years and above), gender, ethnicity (Chinese, Malay, Indian, or Others), marital status (never married, married, divorced/separated, or widowed), educational level (primary and below, secondary, vocational institute, pre-university/junior college, diploma, or university), employment status (employed, economically inactive, i.e., students, housewives, and retirees, or unemployed), and monthly household income in Singapore dollars (SGD) (below 2000, 2000–3999, 4000–5999, 6000–9999, 10,000 and above).

### 2.4. Statistical Analysis

All estimates were weighted to adjust for oversampling and non-response, and post-stratified for age and ethnicity distributions between the survey sample and the Singapore resident population in 2014. Descriptive analyses were performed to establish the lifetime prevalence of DSM-IV and DSM-5 AUD and describe the sociodemographic profile and clinical characteristics of the study population. Sociodemographic correlates of lifetime DSM-IV/DSM-5 AUD were examined using multivariable logistic regression analysis. Associations between DSM-IV/DSM-5 AUD, chronic conditions, and HRQOL summary scores were examined using multivariable logistic and linear regression after controlling for significant sociodemographic factors. Standard errors (SE) and significance tests were estimated using the Taylor series linearization method. All statistical analyses were performed using the Statistical Analysis Software (SAS) system version 9.3 (Cary, NC, USA).

## 3. Results

Figure 1 shows the prevalence of lifetime DSM-IV and DSM-5 AUD in the population. The lifetime prevalence of DSM-IV AUD was 4.6% (0.5% for dependence and 4.1% for abuse) in the adult population, while the lifetime prevalence of DSM-5 AUD was 2.2% (1.6% mild, 0.4% moderate, and 0.2% severe).

Approximately 70% (*n* = 175) of all DSM-IV abuse cases switched to non-cases according to DSM-5 AUD, and 63.9% (*n* = 22) of all DSM-IV diagnostic orphans switched to mild AUD according to DSM-5. The net effect was a relative reduction of 50% in the lifetime prevalence of AUD when diagnosed by DSM-5 compared to DSM-IV. This reduction was statistically significant (Chi Square = 24.9, *p* value < 0.001).

Table 1 shows the percentage of DSM-5 diagnostic orphans who endorsed the DSM-IV abuse criteria. Table 2 shows the prevalence of lifetime DSM-IV and DSM-5 AUD by sociodemographic and clinical characteristics. The lifetime prevalence of MDD, bipolar disorder, GAD, OCD, suicidality, and nicotine dependence was 19.5%, 7.1%, 3.7%, 9.1%, 20.4%, and 20.1%, respectively, among those with DSM-IV AUD and 23.7%, 13%, 7.2%, 13.2%, 24.3%, and 22.5%, respectively, among those with DSM-5 AUD. The treatment gap was 95.3% and 92.3% among those with DSM-IV and DSM-5 AUD, respectively.

Table 3 shows the sociodemographic correlates of lifetime DSM-IV and DSM-5 AUD. Those aged 50 to 64 years, 65 years and above (vs. 18–34 years), and females (vs. males) had lower odds of both DSM-IV and DSM-5 AUD. Those who had a lower education (primary and secondary versus university) had higher odds of DSM-IV and DSM-5 AUD. Those who were widowed (versus married) had lower odds of DSM-5 AUD, while those who were economically inactive (versus employed) and had a higher income (4000–5999 versus below 2000) had lower odds of DSM-IV AUD.

After adjusting for covariates in the logistic regression analysis, the difference between DSM-IV and DSM-5 AUD remained significant with odds of having DSM-IV AUD being 2.1-times higher than DSM-5 AUD (OR = 2.1; 95% CI 1.7–2.7, *p* value < 0.001). Table 4 shows the clinical correlates of lifetime DSM-IV and DSM-5 AUD. Those who were diagnosed with lifetime MDD, bipolar disorder, OCD, suicidality, and nicotine dependence had significantly higher odds of both DSM-IV and DSM-5 AUD. Those diagnosed with lifetime GAD had significantly higher odds of having DSM-5 AUD.

Appendix A shows the relationship between lifetime DSM-IV and DSM-5 AUD and chronic physical conditions. Those diagnosed with both DSM-IV and DSM-5 AUD were significantly associated with lower mental and higher physical composite summary scores (Appendix A).

## 4. Discussion

The current study is the first that provides information on the prevalence of DSM-5 AUD in Singapore as well as the diagnostic overlap between DSM-IV and DSM-5 AUD using data from a national epidemiological study. Our results indicate that the overall prevalence of AUD declines by 50% when cases are defined by the DSM-5 criteria compared to the DSM-IV criteria. The stringent laws and the excise duty in place to restrict alcohol-related harm in Singapore might have resulted in a low prevalence of alcohol dependence. Moreover, most of the cases of alcohol abuse met only one criterion, and these cases were subsequently excluded because they failed to meet the minimum two-criterion threshold for mild AUD as per the DSM-5 classification. In addition, the exclusion of the symptom criterion of use of alcohol leading to legal problems (the second most-endorsed symptom criterion (24%) by those cases who changed from DSM-IV abuse to DSM-5 non-cases) lowered the prevalence of DSM-5 AUD compared to that of DSM-IV AUD in the population.

The results of our study are similar to those of Slade et al. [12] Their study found a relative reduction of 12% in the lifetime prevalence of AUD when diagnosed by the DSM-5 compared to the DSM-IV. Our results were also in line with a study on college students from Lebanon [22]. The authors found that almost twice as many students met the DSM-IV abuse or dependence criteria than the DSM-5 AUD criteria (65% vs. 35%, respectively). This difference in the prevalence between the DSM-IV and DSM-5 criteria was found to be due to the high percentage of DSM-IV alcohol abuse that was contributed mainly by the hazardous alcohol use criterion, i.e., driving while impaired.

### 4.1. Sociodemographic Correlates of DSM 5 AUD in Singapore

The change in the diagnostic criteria did not have a significant impact on the socio-demographic and clinical correlates of the cases. Younger age, male gender, and lower education were associated with higher odds of both DSM-IV and DSM-5 AUD. The prevalence of alcohol consumption and AUD has been shown to decrease with age across several studies [23,24]. The rise in problematic drinking in early adulthood may be associated with a transition to independence and fewer restrictions as young adults start attending colleges and move out of parental homes [25]. The decline in problematic drinking as one gets older or the “maturing out” of the disorder is attributed to increased responsibilities associated with marriage and parenthood [26]. Several factors may contribute to the gender differences. While some studies suggest that women are risk-averse and avoid behaviors such as smoking and alcohol drinking [27], others suggest that cultural expectations may play a role; for example, problematic drinking in men is significantly associated with masculine norms [28], but it is considered an undesirable trait in women who are seen as mothers and homemakers and subjected to greater social sanctions for drinking [29].

Other studies have demonstrated that educational status influences alcohol consumption patterns and alcohol-related harm, with higher educational attainment being associated with a reduced risk of problematic drinking [30,31,32]. It is possible that those with lower educational attainment may experience greater exposure to other risk factors associated with AUD such as high stress and may have less knowledge about the harms of excessive alcohol use [33,34]. On the other hand, those with higher education may be more aware and receptive towards health messaging and have a better understanding of the health benefits of moderate alcohol use and the negative effects of heavy drinking and, thus, manage their drinking without risking harm.

### 4.2. Clinical and Functional Correlates of DSM 5 AUD in Singapore

DSM-IV and DSM-5 AUD were associated with significant comorbidities in terms of both other mental disorders and chronic physical conditions. We found significant associations between lifetime AUD and MDD, bipolar disorder, OCD, suicidality, and nicotine dependence even after we controlled for sociodemographic characteristics. AUD and other mental disorders might be related in several ways, including common genetic susceptibility [35] and shared psychosocial factors that have a role in the development of these disorders [36]. These associations would also explain the association of AUD with lower scores in the MCS component of HRQOL.

Despite this, more than 90% of DSM-5 AUD cases go untreated in Singapore. Fears of stigmatization and beliefs that treatment is ineffective may explain the lack of AUD treatment in Singapore [37,38]. However, local studies have demonstrated positive treatment outcomes for those seeking treatment for their problematic alcohol use [39]. Reviews and meta-analyses of randomized trials conducted elsewhere have also demonstrated the efficacy of brief screening and intervention in various settings among individuals with harmful/hazardous alcohol use who can be prevented from developing AUD [40,41], an area that needs further implementation in Singapore.

Both DSM-IV and DSM-5 AUD were associated with cardiovascular diseases and chronic pain. A longitudinal study by Whitman et al. [42] reported an association between alcohol abuse and myocardial infarction as well as congestive heart failure. The authors suggested that it could be due to alcohol-related cardiomyopathy or due to the confounding effect of smoking or increased body mass index among those with alcohol abuse. Studies have suggested that people experiencing chronic pain may use alcohol for relief [43]. It has also been observed that withdrawal from chronic alcohol use increases pain sensitivity as a part of alcohol withdrawal syndrome [44].

### 4.3. Are DSM 5 Diagnostic Orphans a Cause for Concern?

Our data indicate that a majority (76%) of DSM-5 diagnostic orphans are not accounted for by the removal of the legal criterion, and this group would also go undetected with the adoption of the DSM-5 criteria. This large group may require targeted intervention if they are shown to have poor clinical and functional outcomes, and this may warrant ongoing screening for DSM-IV abuse alone on top of the DSM-5 criteria. It is important to note, however, that we did not examine if the DSM-5 diagnostic orphan group is associated with poorer functional and clinical outcomes, so it is not possible to conclude that they represent an at-risk group. However, the endorsement of ‘failure to fulfill major roles, drinking in hazardous situations, and social and interpersonal problems’ criteria all implicitly suggest a form of functional loss. Future studies should be aimed at addressing this outstanding question. There are also policy implications with the large group of DSM-5 diagnostic orphans; problematic alcohol use may be seen as less prevalent and thus less resources may be dedicated to combating its ills.

## 5. Conclusions

In summary, the strengths of the current study include the use of a large multi-ethnic national sample as well as a standardized diagnostic instrument. The study was among the first to examine the association of DSM5-AUD with other mental and physical conditions as well as HRQOL. However, these results need to be interpreted within the context of some limitations. We used the lifetime perspective instead of a 12-month given the low prevalence of the disorder. The responses were based on self-report and, therefore, subject to recall bias. The main intent of the study was to examine the impact of changing the definitions on cases defined according to the DSM-5 criteria versus the DSM-IV criteria; thus, we did not conduct any validation exercise of the DSM-5 criteria.

Our study highlights the significant reduction in the prevalence of AUD when switching from the DSM-IV to DSM-5 criteria in a population where alcohol consumption, AUD, and severity of AUD are low. While using the DSM-5 criteria would result in a lower prevalence of AUD in Singapore, it remains a highly comorbid condition associated with poor health-related quality of life that is largely untreated, which makes it a public health concern.

## Figures and Tables

**Figure 1 ijerph-20-00285-f001:**
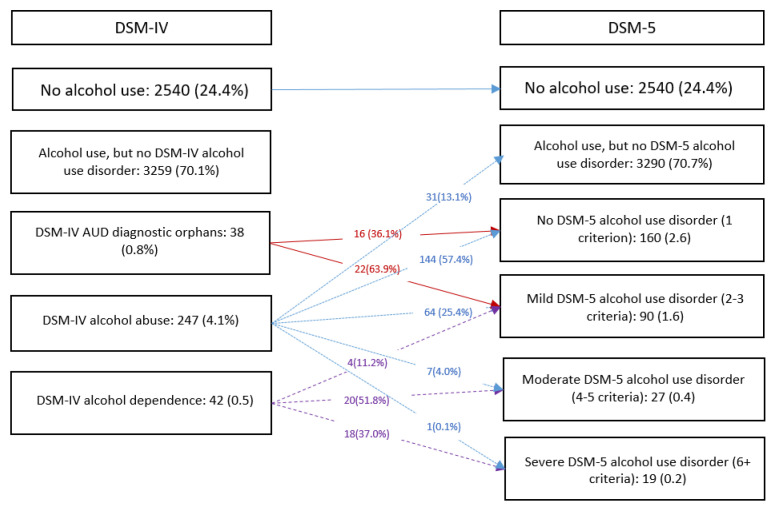
The lifetime prevalence of alcohol use disorders in the general population according to the DSM-IV and DSM-5 criteria (*n* = 6126).

**Table 1 ijerph-20-00285-t001:** DSM-5 diagnostic orphans’ endorsement of DSM-IV abuse criteria (*n* = 175).

	*n*	%
**DSM-IV abuse criteria** Drinking interfered with your work	75	42.9
Drinking resulted in legal problems	42	24.0
Drank in situations where you could get hurt	39	22.3
Drinking caused social or interpersonal problems	30	17.1

**Table 2 ijerph-20-00285-t002:** The lifetime prevalence of DSM-IV and DSM-5 alcohol use disorder by sociodemographic and clinical characteristics.

	DSM-IV		DSM-5	
	*n*	%	*n*	%
**Age group**				
18–34	131	7.2	71	3.8
35–49	81	5.5	36	2.3
50–64	54	2.6	20	1.2
65+	23	1.2	9	0.5
**Gender**				
Male	235	7.5	106	3.5
Female	54	1.9	30	1.0
**Ethnicity**				
Chinese	74	4.4	36	2.1
Malay	75	4.4	38	2.4
Indian	105	6.3	49	3.0
Others	35	6.5	13	2.4
**Marital**				
Never married	107	6.2	59	3.5
Married	156	4.0	67	1.7
Divorced/separated	21	6.1	9	2.5
Widowed	5	1.0	1	0.1
**Education**				
Primary and below	37	4.0	17	2.2
Secondary	80	4.5	43	2.7
Pre-U/Junior College	5	1.2	5	2.0
Vocational/ITE	46	10.0	21	2.9
Diploma	54	5.2	24	2.5
University	67	4.4	26	1.8
**Employment**				
Employed	236	5.4	107	2.5
Economically inactive	31	1.6	16	1.2
Unemployed	22	7.7	13	3.7
**Household Income (SGD/month)**				
Below 2000	55	5.1	40	2.4
2000–3999	70	5.3	34	2.0
4000–5999	43	2.7	30	2.3
6000–9999	52	5.8	21	3.0
10,000 and above	50	4.7	15	1.8
**DSM-IV mental disorders**				
MDD	47	19.5	29	23.7
Bipolar disorder	28	7.1	22	13.0
GAD	16	3.7	13	7.2
OCD	28	9.1	20	13.2
**Treatment gap**	273	95.3	125	92.3
**Suicidality**	58	20.4	36	24.3
**Nicotine dependence**	62	20.1	33	22.5

**Table 3 ijerph-20-00285-t003:** Sociodemographic correlates of lifetime DSM-IV and DSM-5 alcohol use disorder.

	DSM-IV			DSM-5		
	AOR	95% CI	*p* Value	AOR	95% CI	*p* Value
**Age group**						
18–34	Ref.			Ref.		
35–49	0.7	0.4–1.2	0.219	0.6	0.3–1.2	0.145
50–64	0.2	0.1–0.4	<0.001	0.2	0.1–0.5	0.001
65+	0.1	0.03–0.3	<0.001	0.1	0.01–0.4	0.002
**Gender**						
Male	Ref.			Ref.		
Female	0.3	0.2–0.4	<0.001	0.3	0.2–0.6	<0.001
**Ethnicity**						
Chinese	Ref.			Ref.		
Malay	0.7	0.5–1.1	0.155	0.9	0.5–1.5	0.644
Indian	1.3	0.9–1.8	0.206	1.4	0.9–2.2	0.164
Others	1.8	1.1–3.0	0.029	1.3	0.6–2.9	0.530
**Marital**						
Married	Ref.			Ref.		
Never married	1.0	0.6–1.8	0.981	1.3	0.7–2.6	0.409
Divorced/separated	1.6	0.7–3.7	0.237	1.4	0.5–4.1	0.567
Widowed	0.7	0.1–4.1	0.715	0.1	0.01–0.8	0.028
**Education**						
University	Ref.			Ref.		
Primary and below	4.0	1.8–9.3	0.001	5.7	1.8–17.9	0.003
Secondary	2.3	1.2–4.4	0.011	3.6	1.5–8.6	0.005
Pre-U/Junior College	0.5	0.1–2.5	0.429	1.9	0.4–7.9	0.389
Vocational/ITE	2.7	1.4–5.3	0.005	1.6	0.6–3.8	0.339
Diploma	1.5	0.8–2.7	0.186	1.2	0.5–3.1	0.672
**Employment**						
Employed	Ref.			Ref.		
Economically inactive	0.4	0.2–0.9	0.030	0.4	0.2–1.0	0.060
Unemployed	1.4	0.7–3.0	0.358	1.4	0.5–3.7	0.489
**Household Income (SGD/month)**						
Below 2000	Ref.			Ref.		
2000–3999	0.7	0.4–1.3	0.272	0.6	0.3–1.2	0.151
4000–5999	0.4	0.2–0.7	0.005	0.8	0.4–1.7	0.502
6000–9999	0.9	0.5–1.9	0.861	1.2	0.5–3.0	0.753
10,000 and above	0.9	0.4–2.0	0.855	0.8	0.3–2.2	0.700

Note: Adjusted Odds Ratio (AOR) was generated using multivariable logistic regression analyses after controlling for all sociodemographic variables.

**Table 4 ijerph-20-00285-t004:** Clinical correlates of DSM-IV and DSM-5 alcohol use disorder.

	DSM-IV AUD				DSM-5 AUD	
	AOR	95% CI	*p* Value	AOR	95% CI	*p* Value
No MDD	Ref.					
MDD	4.1	2.4–7.0	<0.001	4.7	2.4–9.0	0.001
No Bipolar disorder	Ref.					
Bipolar disorder	5.3	2.5–11.2	<0.001	10.1	4.5–22.7	<0.001
No GAD	Ref.					
GAD	2.3	0.9–5.8	0.073	5.0	1.9–13.2	0.001
No OCD	Ref.					
OCD	2.6	1.4–5.1	0.003	3.7	1.7–8.0	0.001
No suicidality	Ref.					
Suicidality	3.1	1.9–5.2	<0.001	3.3	1.7–6.4	<0.001
No nicotine dependence	Ref.					
Nicotine dependence	5.8	3.5–10.1	<0.001	5.6	2.8–11.1	<0.001

Note: Adjusted Odds Ratio (AOR) was generated from multiple logistic regression after controlling for significant sociodemographic factors identified in the analysis.

## Data Availability

The data is not publicly available due to ethical restrictions.

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
