# Peer review of "Differences in the Prevalence and Profile of DSM-IV and DSM-5 Alcohol Use Disorders—Results from the Singapore Mental Health Study 2016"

_ijerph, 2022, doi:10.3390/ijerph20010285_

Round 1

Reviewer 1 Report

Comments and Suggestions for Authors

In the abstract, please mention major statistical analysis (in method, e.g., logistic regression) and results of logistic regression analysis (e.g., AOR) in results.

In main text, the abbreviation of alcohol use disorders, AUD, was used first in line 43. However, the full nation was used in multiple places. Please use the abbreviation.

In the abstract, Table 1, and text, the prevalence of AUD was 4.7% and this was sum of 0.5% for dependence and 4.1% for abuse. However, sum of 0.5% and 4.1% is 4.6%. If this is simply from rounding calculation (first decimal place), please match the overall prevalence to 4.6% or prevalence of dependence/abuse. This is the same for DSM-5 1.6% + 0.4% + 0.2 = 2.2%, but you reported 2.3%.

Table 1 is confusing. It would be better to report the sum of all numbers in one column is total sample (6,126) and 100%. Remove “Any DSM-IV alcohol use disorder” from the table and mention it in footnote.

DSM-IV

DSM-5

Also, I would separate DSM-IV and DSM-5 into different column. If it is difficult to present in Table format, I would create figure instead of Table. Table S1 also does not necessary.

No alcohol use: 2,540 (24.4%)

No alcohol use: 2,540 (24.4%)

16 (36.1%)

Any AUD

If you keep table format, in Table S1, please separate No DSM-IV alcohol use disorder into two: “No alcohol use” and “Alcohol use, but no DSM-IV alcohol use disorder” as you used in Table 1.

In Table 2, you mentioned (n=175) in the Table title but sum of all numbers is 186. Check the numbers. I am not sure Table 2 is necessary/critical for your research question.

The group differences in Table 3 should be examined with chi-square tests. For example, you mentioned “The lifetime prevalence of MDD, bipolar disorder, GAD, OCD, suicidality and nicotine dependence was higher among those with DSM-5 AUD as compared with DSM-IV AUD” in lines 195-197. Was the group difference in AUD across mental disorders statistically significant in both DSM-IV and DSM-5?

Author Response

5 December 2022

Ref: Differences in the prevalence and profile of DSM‑IV and DSM‑5
alcohol use disorders – Results from the Singapore Mental Health Study 2016.

Manuscript ID: ijerph-1996122

We would like to thank the reviewers for their in-depth and thorough review of our article.

Please find our replies enclosed in bold for easy reference. We have also made the changes to the manuscript as suggested.

 Independent Reviewer(s)' Comments to Author:
Reviewer 1
In the abstract, please mention major statistical analysis (in method, e.g., logistic regression) and results of logistic regression analysis (e.g., AOR) in results.

We have amended the abstract to include the statistical analysis. However if we include the details of all the AORs the abstract will become very long. Therefore, we have highlighted only associations that were different between DSM-IV and DSM-5 AUD.

  1. In main text, the abbreviation of alcohol use disorders, AUD, was used first in line 43. However, the full nation was used in multiple places. Please use the abbreviation.

We apologise for the error and have corrected the same.

  1. In the abstract, Table 1, and text, the prevalence of AUD was 4.7% and this was sum of 0.5% for dependence and 4.1% for abuse. However, sum of 0.5% and 4.1% is 4.6%. If this is simply from rounding calculation (first decimal place), please match the overall prevalence to 4.6% or prevalence of dependence/abuse. This is the same for DSM-5 1.6% + 0.4% + 0.2 = 2.2%, but you reported 2.3%.

The reviewer is right in pointing out that the difference has occurred due to the rounding calculation. We agree with their suggestion and have changed the numbers to 4.6% and 2.2% respectively.

  1. Table 1 is confusing. It would be better to report the sum of all numbers in one column is total sample (6,126) and 100%. Remove “Any DSM-IV alcohol use disorder” from the table and mention it in footnote.

Also, I would separate DSM-IV and DSM-5 into different column. If it is difficult to present in Table format, I would create figure instead of Table. Table S1 also does not necessary.

  1. If you keep table format, in Table S1, please separate No DSM-IV alcohol use disorder into two: “No alcohol use” and “Alcohol use, but no DSM-IV alcohol use disorder” as you used in Table 1.

We thank the reviewer for the suggestion and have made the changes as suggested. Table 1 has been replaced by Figure 1 and S1 has been removed.

  1. In Table 2, you mentioned (n=175) in the Table title but sum of all numbers is 186. Check the numbers. I am not sure Table 2 is necessary/critical for your research question.

The total number of respondents is 175, but some could have endorsed more than one criterion where one of the criteria endorsed was that of ‘Legal problems’. We would like to retain the table as we feel that it shows the spread of the diagnostic orphans by DSM-IV endorsed criteria.

  1. The group differences in Table 3 should be examined with chi-square tests. For example, you mentioned “The lifetime prevalence of MDD, bipolar disorder, GAD, OCD, suicidality and nicotine dependence was higher among those with DSM-5 AUD as compared with DSM-IV AUD” in lines 195-197. Was the group difference in AUD across mental disorders statistically significant in both DSM-IV and DSM-5?

We apologise for the incorrect phrasing. Our intent was just to provide a description of the prevalence of various mental disorders in each group. We have changed the sentence accordingly.

  1. I don’t see any specific reason for separating Tables 4 and 5. Did you run the logistic regression model including all independent variables included in Table 3? Why you separate Tables 4 and 5? Why was the suicidality removed from the regression model in Table 5? Please combine Tables 4 and 5 and report OR. One minor comment: change OR to AOR (adjusted odds ratio) and remove p-values for OR.

We would like to keep the two tables separate as in Table 4 we included all the socio-demographic variables included in Table 3. Table 5 which includes suicidality and nicotine dependence along with the mental disorders was adjusted only for the socio-demographic factors that were significant as shown in Table 4.

We have used the term AOR as suggested by the reviewer.

  1. Also, in Table 4, one group of each independent variable is a reference group. For gender variable (binary indicator), male is a reference group. If MDD, Bipolar, GAD, and OCD are binary indicator (Yes/No), please mention a reference group (No) as you did in gender variable.

We have made the change as suggested by the reviewer.

  1. In Tables 2 and 4, you may use a consistent way to present independent variables. For example, boldface is used to represent variable name (e.g., Age group) and normal font for the category of independent variables (e.g., 18-34 / 35-49 / 50-64 / 65+). However, “DSM-IV mental disorder” is not the name of single variable and MDD, Bipolar, GAD, and OCD are the name of variables. Again, please use a consistent format.

We apologise for the error and have corrected it.

In discussion, you concluded 50% decline of the AUD prevalence. Did you examine if this reduction is statistically significant? Although this is your primary research question, you did not conduct any analysis to examine this. For example, you may check a chi-square test result with a 2 (yes/no in DSM-IV) X 2 (yes/no in DSM-5) table, and then conduct a multiple logistic regression analysis to see the AOR for AUD in DSM-5 (compared to DSM-IV).

We apologise for not providing the information and thank the reviewer for pointing it out to us. We have added this to the Results section as suggested.

One additional minor comment:

I don’t know the education system in Singapore. In general, researchers exclude younger adults, specifically 18-20, when they include education level as a covariate as some of them may not have time to go to college. Their education level might be high school or lower in this study. This bias may be considered.

The education system in Singapore is complex as men must undergo a compulsory National Service before they enter college or after they finish their diploma. Also, respondents may choose to either drop out of school or college or defer their college. Hence, we would not like to exclude any age group from the analyses.

We hope that we have addressed all the comments adequately.

Regards

Authors

Reviewer 2 Report

Comments and Suggestions for Authors

I would just cut out, in the conclusions section, the statements about the therapeutic issues, which are not required. The aim of the study is to confront the two diagnostic models, and results do not imply anything themselves on therapeutic grounds.

Author Response

5 December 2022

Ref: Differences in the prevalence and profile of DSM‑IV and DSM‑5
alcohol use disorders – Results from the Singapore Mental Health Study 2016.

Manuscript ID: ijerph-1996122

We would like to thank the reviewers for their in-depth and thorough review of our article.

Please find our replies enclosed in bold for easy reference. We have also made the changes to the manuscript as suggested.

Reviewer 2

I would just cut out, in the conclusions section, the statements about the therapeutic issues, which are not required. The aim of the study is to confront the two diagnostic models, and results do not imply anything themselves on therapeutic grounds.

We have deleted the sections related to therapeutic issues as suggested by the reviewer.

We hope that we have addressed all the comments adequately.

Regards

Authors